# Vaccination of yearling horses against poly-*N*-acetyl glucosamine fails to protect against infection with *Streptococcus equi* subspecies *equi*

Noah D. Cohen[ID]^[1]*, Colette Cywes-Bentley[2], Susanne M. Kahn[1], Angela I. Bordin[ID][1], Jocelyne M. Bray[1], S. Garrett Wehmeyer[1], Gerald B. Pier[ID][2]*

1 Equine Infectious Disease Laboratory, College of Veterinary Medicine & Biomedical Sciences, Texas A&M University, College Station, TX, United States of America, 2 Harvard Medical School, Brigham & Women's Hospital, Boston, MA, United States of America

* ncohen@cvm.tamu.edu (NDC); gpier@bwh.harvard.edu (GBP)

## Abstract

Strangles is a common disease of horses with worldwide distribution caused by the bacterium *Streptococcus equi* subspecies *equi* (SEE). Although vaccines against strangles are available commercially, these products have limitations in safety and efficacy. The microbial surface antigen β 1→6 poly-*N*-acetylglucosamine (PNAG) is expressed by SEE. Here we show that intramuscular (IM) injection alone or a combination of IM plus intranasal (IN) immunization generated antibodies to PNAG that functioned to deposit complement and mediate opsonophagocytic killing of SEE *ex vivo*. However, immunization strategies targeting PNAG either by either IM only injection or a combination of IM and IN immunizations failed to protect yearling horses against infection following contact with infected horses in an experimental setting. We speculate that a protective vaccine against strangles will require additional components, such as those targeting SEE enzymes that degrade or inactivate equine IgG.

## 1. Introduction

Strangles is an ancient disease of horses that is caused by infection with *Streptococcus equi* subspecies *equi* (SEE) [1–5]. Currently available vaccines against SEE have limitations including the potential to cause disease in vaccinated horses [6–8], the potential to trigger immune-mediated sequelae such as vasculitis or myositis [9–13], lack of high-quality evidence of efficacy, and inability to differentiate natural infection from vaccination (DIVA). Thus, there is great need to identify improved vaccines for strangles.

The microbial surface polysaccharide β 1→6 poly-*N*-acetylglucosamine (PNAG) is evolutionarily conserved and found on the surface of many bacteria, including SEE and *Streptococcus equi* subspecies *zooepidemicus* [14–16]. Structural, genetic and immunochemical studies show that PNAG is highly conserved with only some (10–30%) variability in the level of

**Data Availability Statement:** Data are all contained within the manuscript or supporting information files.

**Funding:** This work was supported by grants from the Grayson-Jockey Club Research Foundation (IM vaccination portion; no grant identifier) and the Morris Animal Foundation (IM+IN vaccination; grant number (D19EQ-012). Additional support was provided by the Link Equine Research Endowment (no grant identifier). The funders had no role in study design, data collection and analysis, decision to publish, or preparation of the manuscript.

**Competing interests:** Gerald B. Pier is an inventor of intellectual properties [human monoclonal antibody to PNAG and PNAG vaccines] that are licensed by Brigham and Women's Hospital to Alopexx Vaccine, LLC, and Alopexx Pharmaceuticals, LLC, entities in which GBP also holds equity. As an inventor of intellectual properties, GBP also has the right to receive a share of licensing-related income (royalties, fees) through Brigham and Women's Hospital from Alopexx Pharmaceuticals, LLC, and Alopexx Vaccine, LLC. GBP's interests were reviewed and are managed by the Brigham and Women's Hospital and Partners Healthcare in accordance with their conflict of interest policies. Colette Cywes-Bentley is an inventor of intellectual properties [use of human monoclonal antibody to PNAG and use of PNAG vaccines] that are licensed by Brigham and Women's Hospital to Alopexx Pharmaceuticals, LLC. As an inventor of intellectual properties, CC-B also has the right to receive a share of licensing-related income (royalties, fees) through Brigham and Women's Hospital from Alopexx Pharmaceuticals, LLC. The remaining authors have declared no competing interests exist. This does not alter our adherence to PLOS ONE policies on sharing data and materials.

acetylation of the amino group on the glucosamine monomer components [14–16]. Antibodies to PNAG occur naturally in many species of animals but are usually neither opsonic nor protective because they cannot facilitate complement deposition and therefore bacterial killing [17–19]. The antibodies induced by a chemically modified glycoform of PNAG, achieved by altering the acetylation pattern to produce deacetylated or **dPNAG**, readily elicits antibodies that recognize native PNAG and provide protection against pathogens expressing native PNAG [16, 20, 21]. Similarly, a synthetic oligosaccharide composed of only 5 monomer units of β-1-6-linked glucosamine (**5GlcNH$_2$**) conjugated to tetanus toxoid (**TT**) as a carrier induces protective immunity against PNAG-producing pathogens [18, 21, 22].

Our laboratories have demonstrated that immunizing pregnant mares against 5GlcNH$_2$ during late gestation protects their foals against pneumonia caused by the bacterium *Rhodococcus equi* [14]. The expression of PNAG by SEE [15] and our positive results against *R. equi* indicated that vaccination targeting PNAG might also protect horses against infection with SEE. In this report we demonstrate that although PNAG is expressed on the surface of SEE, it failed to protect horses when administered either intramuscularly (IM) or when administered both IM and intranasally (IN).

## 2. Materials and methods

### 2.1. Ethics statement

All procedures for this study were reviewed and approved by the Texas A&M Institutional Animal Care and Use Committee (protocol number AUP# IACUC 2015–0335, AUP# IACUC 2017–0293, and AUP# IACUC 2018–0349) and the University Institutional Biosafety Committee (permit number IBC2017-105). The horses used in this study were university-owned, and permission for their use was provided in compliance with the Institutional Animal Care and Use Committee procedures. No horses died or were euthanized as a result of this study.

### 2.2. Overall study design

The study included 4 phases. First, we studied the safety and immunogenicity of intramuscular (IM) administration of PNAG vaccine in 30 horses, including 20 horses that were followed for up to 27 weeks to assess duration of immunity. Second, we conducted a challenge study in 9 yearlings (6 vaccinated with PNAG and 3 sham vaccinated controls). Third, we evaluated the ability of a combination of IM and intra-nasal (IN) vaccination with PNAG to generate antibodies in the serum, nasal passages, and guttural pouches of horses. Fourth, we evaluated the ability of the combination of IN plus IM vaccination to protect yearling horses against SEE. A summary of the phases is listed below.

**Phase 1: *Intramuscular (IM) Safety & Immunogenicity (n = 30 mature horses).*** 20 horses vaccinated IM + 10 unvaccinated controls monitored for 27 weeks for clinical signs, and serum and nasal antibody responses (IgG, IgA, IgG$_1$, IgG$_{4/7}$).

**Phase 2: *IM Contact-Challenge Infection Study (n = 9 yearling horses).*** 6 horses vaccinated IM + 3 controls sham vaccinated, then challenged with contact to infected horse, monitored for antibody responses to vaccine in serum and nasal secretions prior to challenge, and clinical signs of disease subsequent to challenge.

**Phase 3: *IM+Intranasal (IN) Immunogenicty (n = 9 yearling horses).*** 6 horses vaccinated IM+IN and 3 control horses, monitored for serum, intranasal, and guttural pouch antibody responses to vaccination.

**Phase 4: *IM+IN Contact-Challenge Infection Study (n = 5 yearling horses).*** 3 IM+IN vaccinated horses, and 1 unvaccinated control horse contact-challenged with 1 horse infected with SEE.

## 2.3. Vaccine

Horses in the safety and immunogenicity study receiving an IM vaccination were injected with 200 μg of a vaccine composed of synthetic pentamers of β 1→6-linked glucosamine conjugated to tetanus toxoid (5GlcNH$_2$TT; ratio of oligosaccharide to protein 35–39:1; Product AV0328, Alopexx Enterprises, LLC, Concord, MA) diluted to 900 μl in sterile medical grade physiological (*i.e.*, 0.9% NaCl) saline solution (PSS) combined with 100 μl of Specol (Stimune® Immunogenic Adjuvant, Prionics, Lelystad, Netherlands, now part of Thermo-Fischer Scientific), a water-in-oil adjuvant [14]. Horses in the unvaccinated group were sham injected with an equivalent volume (1 ml) of sterile PSS; no adjuvant was administered to control horses. Horses were randomly assigned to the vaccine group (n = 20) or the saline vaccinated control (n = 10). These horses were vaccinated and then boosted 3 weeks later; controls were administered saline at these same times.

To examine whether a combination of intranasal (IN) and IM vaccination could generate antibody responses in the upper airways, we vaccinated 3 horses IM only, 3 horses IN only, and 3 horses IM + IN. For these purposes, we used the adjuvant Montanide™ Gel 01 (Seppic, Cedex, France; 100 microliters [10% volume]) for IN administration and 125 microliters [12.5%] for IM administration) and 200 μg of the same PNAG pentamers described above. Administration of IN vaccinations was performed using the intranasal canulae from a commercially-available vaccine (Pinnacle® IN, Zoetis, Kalamazoo, Michigan, USA).

## 2.4. Horses and specimen collection

All horses used for this study had titers against the M protein of SEE (SeM) < 1:800. Thirty mature Quarter Horses (QH) of both sexes (10 males and 20 females) of ages from 5 to 19 years were used to evaluate the safety and immunogenicity of the PNAG vaccine. Eighteen yearling QH of both sexes (8 geldings and 10 females) were used for the IM challenge study. Nine yearlings were used for the IM+IN immunogenicity study (4 geldings and 5 females), and 5 yearling horses (3 males and 2 females) were used for the pilot study of efficacy of IM +IN vaccination. All horses were owned by Texas A&M University except for 12 yearlings that were leased for the IM+IN vaccination project.

For the safety and immunogenicity study of IM vaccination, horses were monitored by physical examination twice daily to observe clinical signs including local and systemic reactions. Monitoring began 1 week prior to vaccination (or booster) and was continued daily until 1 week after the booster (6 weeks total). Complete blood counts (CBCs) and serum biochemistry profiles were performed weekly from 1 week prior to vaccination until 1 week after vaccination. For purposes of analysis, concentrations of red blood cells (RBCs), white blood cells (WBCs), neutrophils, total globulins, fibrinogen, creatinine, and aspartate aminotransferase (AST) were compared between groups. Blood was collected for CBCs and serum biochemical analysis by jugular venipuncture (16 ml total volume).

For immunogenicity studies, serum samples were collected immediately prior to vaccination, prior to boosting 3 weeks later, and 2, 4, 8, 12, 24 weeks after the second booster injection (hereafter referred to as weeks 0, 3, 5, 7, 11, 15, and 27). In addition, nasal secretions were collected at weeks 0, 3, 5, 7, 11, and 15 using sterile absorbent foam plugs (Identi-Plugs Size B2, Jaece Industries, North Tonawanda, NY, USA) tethered to a string (**S1 Fig**). Horses were sedated with xylazine (0.5 mg/kg), and the sponges were inserted into the nasal passages of each horse to a depth of approximately 5 to 8 cm using a plastic equine insemination catheter as an introducer, and allowed to sit in place for 5 minutes to collect nasal secretions. After 5 minutes, sponges were retrieved by pulling on the string, inserted into conical tubes, placed in a cooler with cold packs, and returned to the laboratory. Sponges were secured to the top of

the conical tube using the string, and centrifuged for 10 minutes at 600 x g to collect nasal secretions for serologic testing.

To evaluate immunogenicity of IM + IN vaccination, serum and nasal sponges were collected as described above. In addition, guttural pouch lavage was performed using endoscopic guidance and sterile, 2.4-mm diameter transendoscopic aspiration catheters (PW-B 2423, Endoscopy Support Services, Brewster, NY, USA) passed via the biopsy channel of the endoscope. For each horse, the left guttural pouch was lavaged with 60 ml of sterile PSS, and this fluid was recovered from the guttural pouch by aspiration. The volume of guttural pouch lavage aspirated was recorded. These samples were tested by ELISA for IgG and IgA antibodies recognizing PNAG, C1q deposition onto PNAG antigen, and opsonophagocytic killing of SEE.

## 2.5. Challenge model

To evaluate the efficacy of IM vaccination with PNAG to protect horses against strangles, we used 6 vaccinated yearlings (3 male and 3 female) and 3 unvaccinated controls (2 male and 1 female). Each vaccinated horse was injected twice, 3 weeks apart, with the $5GlcNH_2$-TT conjugate vaccine; control horses were injected with an equivalent volume (1 ml) of sterile PSS. Serum samples and nasal sponges were collected from all horses prior to vaccination and immediately prior to infection to document immune responses to PNAG in vaccinates and controls. A contact challenge model was used: yearling horses were housed in pens of 3 (2 pens of 3 vaccinates, and 1 pen with 3 control horses), with at least 20 m separation between pens harboring vaccinates and controls. In each pen, a mature horse with strangles was introduced; these horses were university-owned horses from a concurrent outbreak [8]. The affected mature horses were confirmed to be culture-positive for SEE, and had clinical signs including fever, copious nasal discharge, and submandibular and retropharyngeal lymphadenitis. The pens in which each 4 horses (3 yearlings and 1 infected horse) were approximately 1 acre and had a single water trough. Personal protective equipment and isolation procedures were used to avoid cross-contamination by personnel between the different groups of horses.

To evaluate the efficacy of IM+IN vaccination, we infected a yearling horse IN with a dose of 2.5 x $10^7$ colony forming units (CFU) of SEE in a volume of 1 ml in each nostril (total dose, 5 x $10^7$ CFU) via an intranasal catheter. This infected horse was then housed with 3 yearlings that have been vaccinated and boosted IM+IN 3 weeks prior to exposure, and 1 unvaccinated control horse.

## 2.6. ELISA testing for antibody responses

Systemic humoral responses as well as nasal washes and guttural pouch lavages were assessed by quantifying concentrations in serum by ELISA from absorbance values of PNAG-specific total IgG, IgA, or IgG subisotypes $IgG_1$ and $IgG_{4/7}$. ELISA plates (either Immulon 4 HBX 96, Thermo Fisher or Maxisorp, Nalge Nunc International, Rochester, NY, USA) were coated with 0.6 μg/ml of purified PNAG [18] diluted in sensitization buffer (0.04M $PO_4$, pH 7.2) overnight at 4°C. Plates were washed 3 times with PBS containing 0.05% Tween 20, blocked with 120 μl PBS containing 1% skim milk for 1 hour at 37°C, and washed again. Horse serum, nasal and guttural pouch samples were initially diluted in incubation buffer (PBS with 1% skim milk and 0.05% Tween 20) to 1:100 for total IgG titers and IgA titers, and 1:64 for $IgG_1$ and $IgG_{4/7}$ detection. A positive control from a horse previously vaccinated against PNAG and known to have a high titer, along with normal horse serum known to have a low PNAG titer, were used as positive and negative controls, respectively. For the subisotype assays, immune rabbit serum (rabbit anti-$5GLcNH_2$-TT) was diluted to a concentration of 1:102,400 as a positive control

and used as the denominator to calculate the endpoint OD ratio of the experimental OD values. The immune rabbit serum was used to account for inter-plate variability and negative control of normal rabbit serum were included with the equine serum samples. After 1-hour incubation at 37˚C, the plates were washed 3 times as described above. For total IgG titers, rabbit anti-horse IgG whole molecule conjugated to alkaline phosphatase (#A6063, Sigma-Aldrich, St. Louis, MO, USA) was used to detect binding. For IgA titers, rabbit anti-horse IgA conjugated to horseradish peroxidase (Bethyl Laboratories, Montgomery, TX, USA) was used. For IgG subisotype detection, 100 μl of goat-anti-horse $IgG_{4/7}$ (Lifespan Biosciences, Seattle, WA, diluted at 1:90,000 or mouse anti-horse $IgG_1$ (AbD Serotec, Raleigh, NC, USA), diluted at 1:25,000) were added to the wells and incubated for 1 hour at room temperature. After 3 washes, pNPP substrate (1 mg/ml in pH 10 carbonate buffer) for the alkaline phosphatase conjugated secondary antibody was added to the plates for the total IgG titers, while for the peroxidase-conjugated antibody to mouse IgG, SureBlue Reserve One Component TMB Microwell Peroxidase Substrate (SeraCare, Gaithersburg, MD, USA) was added to the wells. Plates were incubated for 15 to 60 minutes at 22˚C in the dark. The reaction was stopped by adding sulfuric acid solution to the wells. Optical densities were determined at 450 nm by using microplate readers. Equine subisotype concentrations of PNAG-specific IgA, $IgG_1$, and $IgG_{4/7}$ were also quantified in nasal secretions using the same protocol described above for serum. Nasal secretion samples were diluted in incubation buffer (PBS with 1% skim milk and 0.05% Tween 20) to 1:100 for IgA, 1:8,192 for $IgG_1$, and 1:4096 for $IgG_{4/7}$ detection. For total IgG endpoint titers were calculated by linear regression using a final $OD_{405}nm$ value of 0.5 to determine the reciprocal of the maximal serum dilution reaching this value. For IgA and IgG subisotypes, an endpoint OD titer was calculated by dividing the experimental OD values with that achieved by the positive control on the same plate.

## 2.7. C1q deposition assays

Serum endpoint titers for deposition of equine complement component C1q onto purified PNAG were determined by ELISA. ELISA plates were sensitized with PNAG and blocked with skim milk as described above, dilutions of different horse sera added in 50 μl-volumes after which 50 μl of 5% intact, normal horse serum was added. After 60 minutes incubation at 37˚C, plates were washed and 100 μl of goat anti-human C1q, which also binds to equine C1q, diluted 1:1,000 in incubation buffer added and plates incubated at room temperature for 60 minutes. After washing, 100 μl of rabbit anti-goat IgG whole molecule conjugated to alkaline phosphatase diluted 1:2,000 in incubation buffer was added and a 1-hour incubation at room temperature carried out. Washing and developing of the color indicator was then carried out as described above, and endpoint titers determined as described above for IgG titers by ELISA.

## 2.8. Opsonic killing assays

To determine opsonic killing of SEE, bacterial cultures were routinely grown overnight at 37˚C on chocolate-agar plates, and then killing assessed using modifications of previously described protocols [14]. Modifications included use of EasySep™ Human Neutrophil Isolation Kits (Stem Cell Technologies Inc., Cambridge, Massachusetts, USA) to purify PMN from blood, and use of gelatin-veronal buffer supplemented with $Mg^{++}$ and $Ca^{++}$ (Boston Bioproducts, Ashland, Massachusetts, USA) as the diluent for all assay components. Final assay tubes contained, in a 400-μl volume, $2 \times 10^5$ human PMN, 10% (final concentration) SEE-absorbed horse serum as a complement source, $2 \times 10^5$ SEE cells and the serum dilutions. Tubes were incubated with end-over-end rotation for 90 minutes then diluted in BHI with 0.05% Tween and plated for bacterial enumeration.

### 2.9. Data analysis for challenge studies and immune responses

The cumulative incidence (and 95% confidence interval) of adverse events was calculated using exact methods. The values of the following hematologic and serum chemistry parameters were compared between times and groups using linear mixed-effects models: concentrations of WBCs, neutrophils, lymphocytes, eosinophils, basophils, monocytes, RBCs, and platelets; PCV; concentration of total serum protein, fibrinogen, globulins, albumin, creatinine, BUN, and bilirubin; and activities of creatinine phosphokinase (CK) and aspartate amino transferase (AST).

The effects of group (vaccinate or control) and time on concentrations of red blood cells (RBCs) white blood cells (WBCs), neutrophils, total globulins, fibrinogen, creatinine, and aspartate aminotransferase (AST) were compared using linear mixed-effects models (nlme package). The proportion of horses that developed clinical signs were compared between groups using Fisher's exact test, and the duration of clinical signs were compared between vaccinates and controls using Wilcoxon rank-sum test.

For comparisons of immune responses, endpoint titers on each day were compared using repeated-measures ANOVA and mixed-effects model and the two-stage linear step-up procedure of Benjamini, Krieger and Yekutieli for multiple comparisons using the PRISM 8.4.3 software package.

## 3. Results

### 3.1. Safety following intramuscular vaccination

Twenty horses vaccinated twice IM and 10 unvaccinated horses (controls) were evaluated for local responses to vaccination, serum chemistry changes and antibody responses. One (1) horse developed swelling in the right side of its neck after vaccination. The estimated proportion of events was 5% (95% CI, <0.1% to 24.9%). The swelling was moderate in size (10 x 10 cm), and resolved gradually over a 2-week period. This event was attributed to improper injection technique (*i.e.*, injection given too low [ventral] in the horse's neck) when the booster was administered; this resulted because the horse's head was down while it was grazing and an investigator inadvertently administered the vaccine in the wrong anatomic location. No other adverse reactions were noted. The horse did not develop any other outward signs of disease (fever, inappetence, pain when the area was palpated).

No significant differences between vaccinates or controls were observed at any of the sampling times for concentrations in blood of RBC, WBC, neutrophils, fibrinogen, globulins, and creatinine or for activity of AST (**S2**–**S8** Figs).

### 3.2. Immune responses following intramuscular vaccination

Immunization with PNAG resulted in significant increases in end-point titers of IgG recognizing PNAG (**Fig 1**), but not in controls. IgG endpoint titers for the 10 controls remained unchanged (**Fig 1A**; Fixed effect (type III) P = 0.1471) whereas there were significant increases in these titers among the vaccinates (**Fig 1B**; Fixed effect (type III) P = 0.0234; P values for multiple comparisons shown in figure). The mean titer differences from day 0 to day 21 was 903 (SE of difference = 285); from day 0 to day 35 (14 days after booster dose) was 5323 (SE of difference = 2009) and between day 0 and day 49 was 3231 (SE of difference = 1264), thus showing a drop in titer between days 35 and 49 (P = 0.0169). The mean increase in IgG titer 14 days following the booster dose was 4420 (SE of difference = 1962) (P = 0.0363).

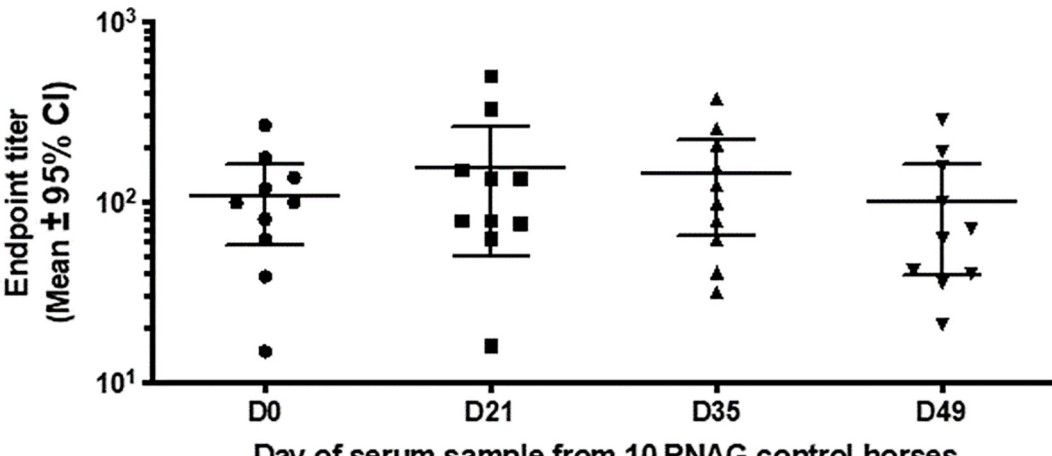

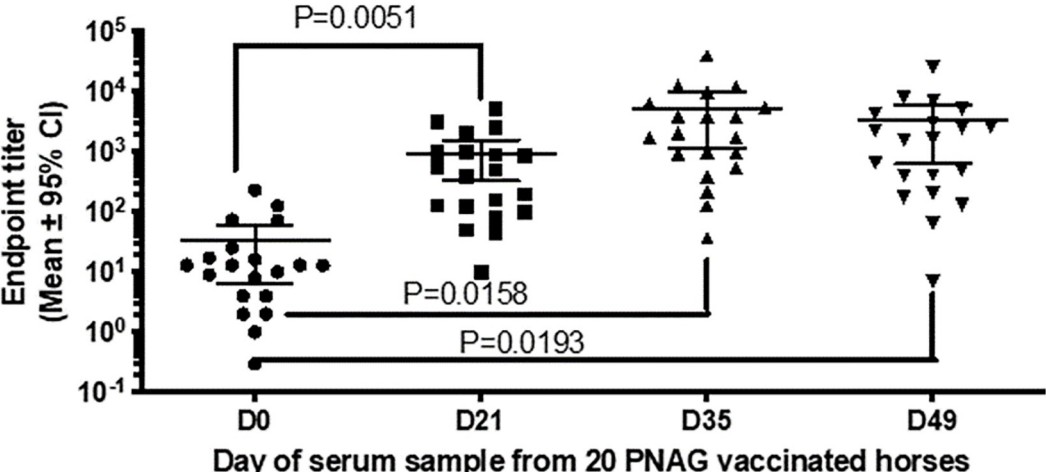

**Fig 1. Immune responses of horses to IM immunization with the PNAG vaccine.** Concentrations (endpoint titers) were unchanged over 49 days in control horses injected with saline (A) but were significantly increased from baseline (D0 = Day 0) at D21 after the initial vaccination, at D35 after a booster dose given at D21, and 14 days (D49) later among 20 PNAG-vaccinated horses (B). P values from multi-group comparisons in fixed-effects model.

Serum concentrations of IgA, $IgG_1$, and $IgG_{4/7}$ were significantly ($P < 0.05$) increased among vaccinates but not controls (**Fig 2A–2C**), and values remained significantly increased through 11 weeks post-vaccination for IgA and $IgG_1$ and 15 weeks for $IgG_{4/7}$.

Concentrations of IgA, $IgG_1$, and $IgG_{4/7}$ in nasal secretions also increased significantly ($P < 0.05$) among vaccinates but not controls (**Fig 3A–3C**), and values remained significantly increased through weeks 7, 11, and 15 post-vaccination for IgA, $IgG_1$, and $IgG_{4/7}$, respectively.

### 3.3. IM vaccination fails to protect yearling horses

Results indicated that all horses had developed serological and nasal mucosal responses to IM vaccination with PNAG prior to challenge. All 9 yearling horses developed clinical signs of strangles (nasal discharge, guttural pouch empyema, and draining submandibular lymphadenitis) irrespective of vaccination status within 16 days of exposure to an infected horse, and there were no significant differences among yearlings with regard to duration of signs (**Table 1**).

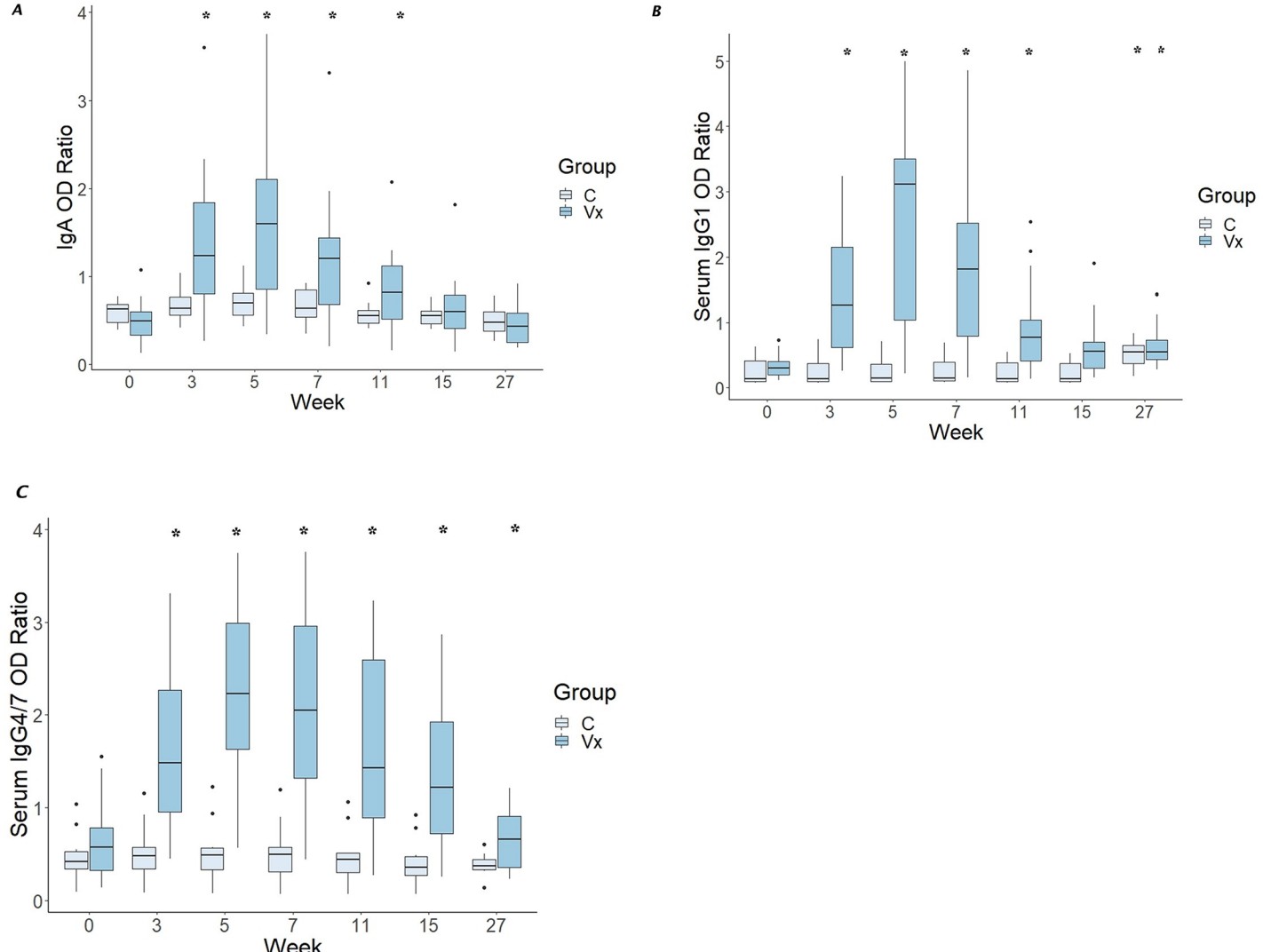

**Fig 2. Vaccination significantly increased serum titers of IgA, IgG$_1$, and IgG$_{4/7}$.** Optical density ratios from serum samples relative to a positive control of IgA (panel A), IgG$_1$ (panel B), and IgG$_{4/7}$ (panel C) antibodies among IM-only vaccinates (Group = Vx; n = 20) and controls (Group = C; n = 10). The bottoms and tops of the boxes represent the 25th and 75th percentiles, and the horizontal lines in the boxes represent the medians; the thin vertical lines extend to a multiple of 1.75 of the respective interquartile distance, and the small circles represent outliers. Asterisks indicate significant (P < 0.05) difference from baseline (Week 0) among vaccinates. For IgA (Panel **A**), concentrations were significantly higher than baseline through week 11 after initial vaccination (12 weeks after booster injection); for controls, there were no significant differences in titers at any of the time point comparisons. For IgG$_1$ (Panel **B**), concentrations were significantly higher than week 0 from weeks 3 through 11, and again at week 27. For controls, values at week 27 were significantly (P < 0.05) higher than for all other times, but no other differences were significant. We attributed the higher titers at week 27 in both groups to be attributable to an artifact of testing or acquisition of natural antibody to PNAG. For IgG$_{4/7}$ (Panel **C**), concentrations were significantly higher than week 0 from weeks 3 through 15; for controls, there were no significant differences among times.

## 3.4. Vaccination IM+IN generates IgG titers to PNAG in guttural pouches and nasal washes

Immunization of yearling horses with 200 μg doses of the PNAG 5GlcNH$_2$-TT vaccine plus Montanide GelA given IM twice three weeks apart along with 2 IN doses given 3 weeks after the first IM dose and again 3 weeks later, elicited increases in IgG in serum, guttural pouch washes and nasal secretions (**Fig 4**). Three horses and a control were initially used to assess titers out to day 59 and subsequently 3 more horses were immunized then exposed to a non-

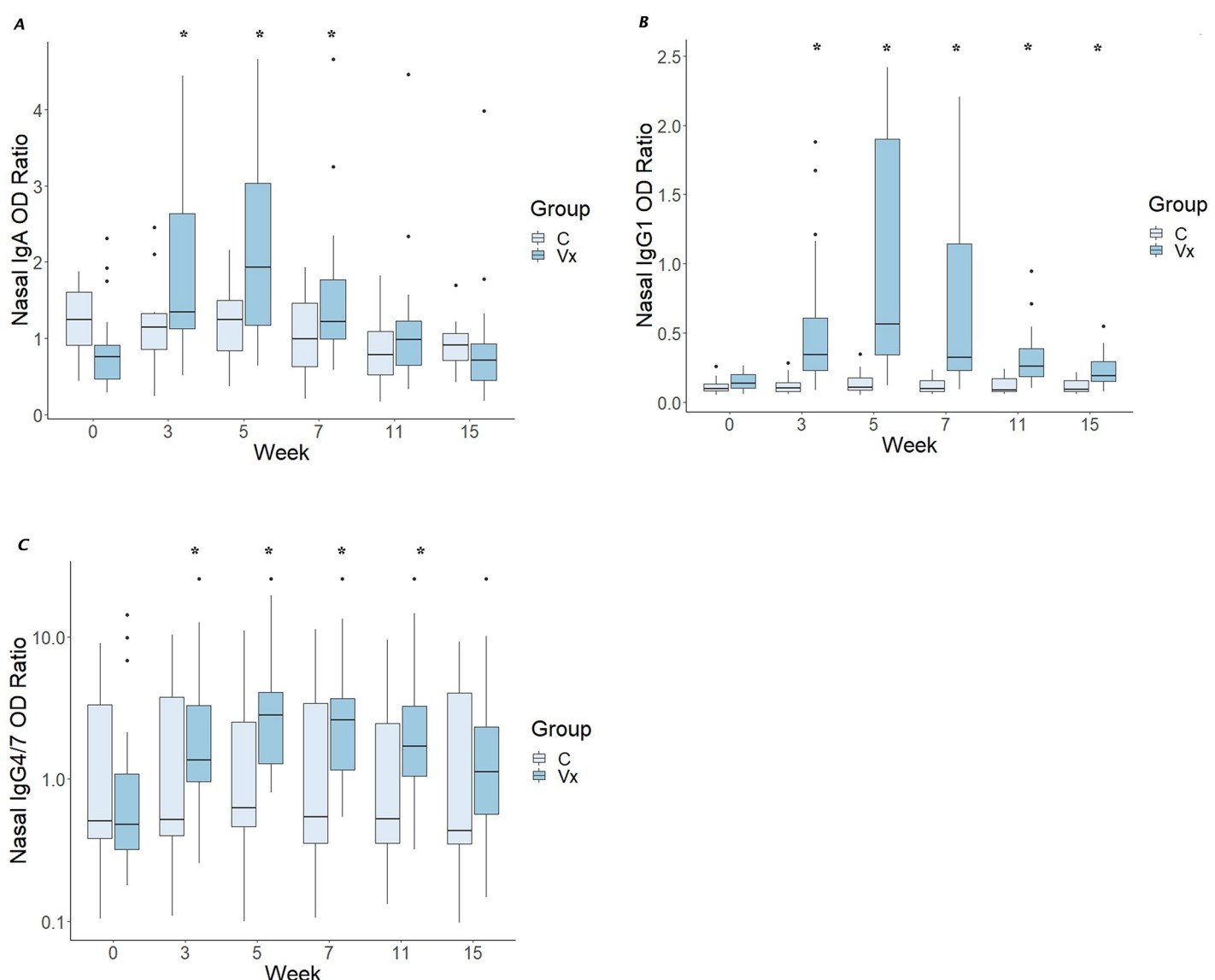

**Fig 3. Vaccination significantly increased nasal titers of IgA, IgG$_1$, and IgG$_{4/7}$.** Boxplot of ratios of optical density of antibodies in nasal samples relative to a positive control of IgA (panel A), IgG$_1$ (panel B; next page), and IgG$_{4/7}$ (panel C) among vaccinates (Group = Vx; n = 20) and controls (Group = C; n = 10). The bottoms and tops of the boxes represent the 25$^{th}$ and 75$^{th}$ percentiles, and the horizontal lines in the boxes represent the medians; the thin vertical lines extend to a multiple of 1.75 of the respective interquartile distance, and the small circles represent outliers. Asterisks indicate significant (P < 0.05) difference from baseline (Week 0) among vaccinates. For controls, there were no significant differences in concentrations among times. For IgA (**A**), IgG$_1$ (**B**), and IgG$_{4/7}$ (**C**), concentrations were significantly (P < 0.05) higher than baseline through week 7, 11, and 15, respectively.

immune, SEE-infected control. Because of the small number of horses in each group no statistical analyses were undertaken.

## 3.5. Opsonic killing of SEE

We evaluated the OPK activity in the sera from a subset of horses immunized IM with the PNAG vaccine and found among 5 of these there were good increases in the killing of SEE (>50% bacteria killed at serum dilution of 10 or greater). No killing of >10% was detected in comparably diluted pre-immunization samples from the horses. However, as all of the

Table 1. Summary of clinical signs in 9 yearling horses (6 PNAG vaccinated and 3 sham-vaccinated yearling controls).

| Variable | Vaccinates (n = 6) | Controls (n = 6) | P value |
|---|---|---|---|
| | Median (range) or Proportion | Median (range) or proportion | |
| Duration exposure to onset (days) | 7 (6 to 16) | 7 (6 to 7) | 1.0000 |
| Duration of clinical signs (days) | 25 (15 to 35) | 24 (21 to 35) | 1.0000 |
| Duration of fever (days) | 10 (8 to 17) | 8 (7 to 10) | 0.3496 |
| Nasal discharge | 100% | 100% | 1.0000 |

immunized horses developed clinical strangles, it was clear this measure of antibody function did not correlate with protection so we discontinued testing of the remaining samples. A similar increase in OPK activity was measured in the horses immunized IM + IN, which also all developed strangles. Determinations of OPK activity in nasal secretions were hampered by the low volumes recovered that prevented testing at a sufficient concentration of the sample in this assay that requires higher antibody concentrations than does the ELISA to measure antibody binding. Guttural pouch lavage fluid had very low OPK activity, likely due to the inability of the sampling procedure to recover antibody from that site at its physiological concentrations and at sufficient concentrations to make a determination of OPK activity.

### 3.6. IM+IN vaccination fails to protect yearling horses

On the basis of findings that IM+IN vaccination increased PNAG titers in the nasal washes and guttural pouches, we used a contact challenge model to determine whether protection was possible in a pilot study. We vaccinated 3 yearling horses IM+IN, and included a negative control horse. We demonstrated that the yearlings developed titers to PNAG in serum, nasal secretions, and guttural pouch fluid (**Fig 4,** symbols that are half-filled squares). A yearling horse infected intra-nasally developed fever (rectal temperature > 39.4˚F) within 24 hours, purulent nasal discharge by 5 days after infection, and signs of retropharyngeal and submandibular swelling by 7 days after infection. All 3 yearling horses vaccinated IM+IN developed clinical signs of strangles (fever, nasal discharge, guttural pouch empyema, and draining submandibular lymphadenitis), as did the 1 unvaccinated control horse. We elected not to pursue planned challenge of additional vaccinated horses on the basis of these negative results.

## 4. Discussion

In this project, we demonstrated that vaccination targeting the conserved surface microbial polysaccharide, PNAG, administered either IM or by a combination of IM and IN immunizations, failed to mediate protection against SEE, a highly important pathogen of horses. These results were unanticipated because vaccination of mares against PNAG protected their foals against intra-bronchial infection with *R. equi* [14]. Initially we expected protection against SEE, as initial analyses by confocal microscopy indicated PNAG was highly expressed on the microbial surface [15], and because our *in vitro* results indicated that antibodies targeting PNAG can mediate opsonophagocytic killing of SEE. However, subsequent to the immunization and challenge studies we found the PNAG epitope was actually expressed on a streptococcal M-like protein, most likely as a small oligosaccharide [15]. This implies that either this type of expression of the PNAG epitope is insufficient to allow for antibody-mediated protection due to its location or conformation on the microbial surface, or other factors expressed by SEE interfered with the protective efficacy of the vaccine-induced antibody.

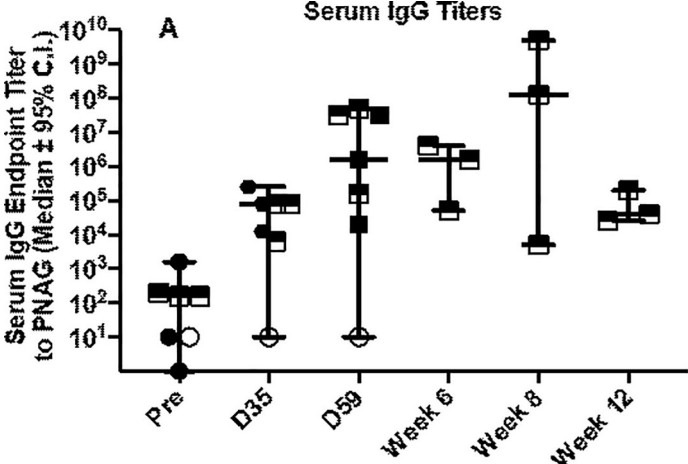

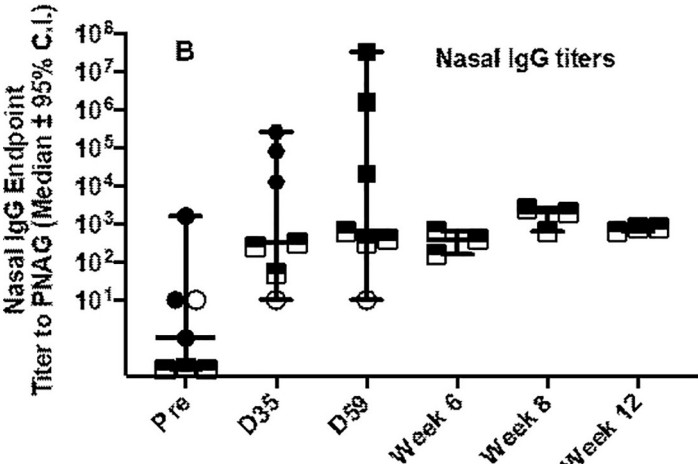

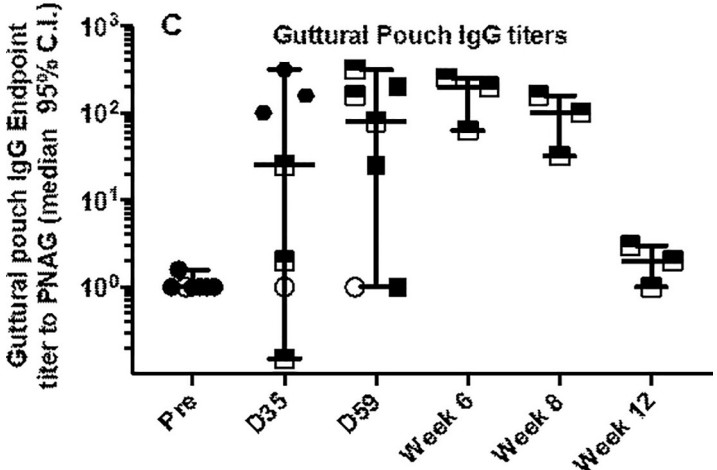

**Fig 4.** A combination of IM and IN immunization induces IgG titers to PNAG in serum (A), nasal secretions (B), and guttural pouch lavages (C). Symbols represent endpoint titers calculated by liner regression. Open circle is control, non-immunized horse. Filled squares represent titers in immunized but not challenged horses. Half-filled squares represent titers in horses challenged with SEE. Samples were only collected from 4 horses used in the initial immunization study through day 59.

As with most failed studies, the reasons for this failure to protect are unclear, but our results indicate it is not because IN vaccination is required. While it is noteworthy that we observed higher titers in guttural pouch lavage fluid after IM+IN vaccination compared with IM vaccination alone, IM+IN vaccination failed to protect yearling horses against challenge. In addition to the manner in which the PNAG-epitope is expressed on SEE noted above, there are other potential explanations for the failure that could be investigated to improve on producing an effective vaccine for strangles. The age of horses studied could have contributed to our negative results. Yearling horses are considered to be highly susceptible to getting strangles, and the modified-live vaccine against strangles that is licensed in North America is labelled to indicate it cannot be safely used in yearling horses. We intentionally selected yearling horses to challenge because we wanted to set a high bar for protection, and because we could accurately determine their having lack of history of exposure to horses with strangles. We did not remove the clinically-affected horses from the pens with the vaccinates or controls, which might have represented a greater exposure to SEE than would occur naturally where, presumably, a horse with clinical signs would be isolated from other horses.

SEE is known to express endopeptidases [23, 24], and it is conceivable these are able to degrade the anti-PNAG antibodies before they are able to bind and attract complement to the bacterial surface. A strangles vaccine candidate that was recently shown to reduce clinical signs of strangles in a challenge study of ponies contained antigens targeting SEE enzymes that degrade IgG or inhibit its functional activity [25]. Targeting these virulence factors could be essential, as Naegeli *et al*. showed that the IgG glycan hydrolase from *S. pyogenes* decreased phagocytic killing of bacteria due to glycan loss and reduction in complement-activation activity, and its expression increased virulence of the organism in a mouse model [26]. Glycan-dependent complement activation by antibody to PNAG is essential for protective efficacy [14, 17], suggesting that antigens inducing neutralizing antibody to the SEE endopeptidase IgeE and the EndoSee glycan hydrolase may need to be included in an effective vaccine. This could also explain the discrepancy between the effectiveness of anti-PNAG antibodies to mediate killing *ex vivo* whilst failing to protect against infection *in vivo*.

This study had a number of limitations. The contact-challenge model does not permit a consistent exposure to each individual horse. Nonetheless, all horses exposed to a horse with strangles in the same pen developed disease. The model we used is likely more severe than natural exposure because horses with clinical signs would generally be isolated from apparently healthy horses in most settings. Perhaps our vaccine would have protected against a less severe challenge. Our sample sizes were purposely small. We had planned to use more horses in both challenge studies, but we felt compelled not to create more disease after observing absence of any evidence of efficacy.

In summary, vaccination of yearling horses against PNAG was safe and stimulated serum and nasal antibodies after IM injection, and more consistently generated antibody responses in the guttural pouches after IM+IN vaccination. Vaccination with PNAG either IM or IM+IN, however, failed to protect yearling horses against strangles. Our results indicate that, although many bacteria express PNAG on their surfaces and that functional anti-PNAG antibodies can protect against infection with *Rhodococcus equi* [14], protection cannot be expected for all bacteria that express PNAG.

## Supporting information

**S1 Fig. Apparatus used for sampling nasal secretions.** Sterile absorbent foam plugs (Identi-Plugs Size B2, Jaece Industries, North Tonawanda, NY, USA) were tethered to a string and were inserted into the nasal passages of each horse to a depth of approximately 5 to 8 cm using a plastic equine insemination catheter as an introducer.
(JPG)

**S2 Fig. Plot of red blood cell (RBC) concentration in blood for control horses (C, n = 10; circle symbols) or vaccinated horses (V, n = 20; triangle symbols) by week.** The 6 vertical panels represent weeks, with week -1 representing sampling 1 week prior to vaccination, and week 0 representing the day of vaccination, followed by weeks 1, 2, 3, and 4 after vaccination. There were no significant differences among groups within times, or within group over time.
(TIF)

**S3 Fig. Plot of white blood cell (WBC) concentration in blood for control horses (C, n = 10; circle symbols) or vaccinated horses (V, n = 20; triangle symbols) by week.** The 6 vertical panels represent weeks, with week -1 representing sampling 1 week prior to vaccination, and week 0 representing the day of vaccination, followed by weeks 1, 2, 3, and 4 after vaccination. There were no significant differences among groups within times, or within group over time.
(TIF)

**S4 Fig. Plot of neutrophil concentration in blood for control horses (C, n = 10; circle symbols) or vaccinated horses (V, n = 20; triangle symbols) by week.** The 6 vertical panels represent weeks, with week -1 representing sampling 1 week prior to vaccination, and week 0 representing the day of vaccination, followed by weeks 1, 2, 3, and 4 after vaccination. There were no significant differences among groups within times, or within group over time.
(TIF)

**S5 Fig. Plot of fibrinogen concentration in blood for control horses (C, n = 10; circle symbols) or vaccinated horses (V, n = 20; triangle symbols) by week.** The 6 vertical panels represent weeks, with week -1 representing sampling 1 week prior to vaccination, and week 0 representing the day of vaccination, followed by weeks 1, 2, 3, and 4 after vaccination. There were no significant differences among groups within times, or within group over time.
(TIF)

**S6 Fig. Plot of concentration of globulins in blood for control horses (C, n = 10; circle symbols) or vaccinated horses (V, n = 20; triangle symbols) by week.** The 6 vertical panels represent weeks, with week -1 representing sampling 1 week prior to vaccination, and week 0 representing the day of vaccination, followed by weeks 1, 2, 3, and 4 after vaccination. There were no significant differences among groups within times, or within group over time.
(TIF)

**S7 Fig. Plot of creatinine concentration in blood for control horses (C, n = 10; circle symbols) or vaccinated horses (V, n = 20; triangle symbols) by week.** The 6 vertical panels represent weeks, with week -1 representing sampling 1 week prior to vaccination, and week 0 representing the day of vaccination, followed by weeks 1, 2, 3, and 4 after vaccination. There were no significant differences among groups within times, or within group over time.
(TIF)

**S8 Fig. Plot of activity of aspartate aminotransferase (AST) in blood for control horses (C, n = 10; circle symbols) or vaccinated horses (V, n = 20; triangle symbols) by week.** The 6

vertical panels represent weeks, with week -1 representing sampling 1 week prior to vaccination, and week 0 representing the day of vaccination, followed by weeks 1, 2, 3, and 4 after vaccination. There were no significant differences among groups within times, or within group over time.
(TIF)

## Acknowledgments

The authors thank Mrs. Ellen Ruth Morris, Ms. Courtney Brake, Alex Lucas, Howard Fisher, and Victor Pineda for technical assistance.

## Author Contributions

**Conceptualization:** Noah D. Cohen, Colette Cywes-Bentley, Gerald B. Pier.

**Data curation:** Noah D. Cohen, Colette Cywes-Bentley, Susanne M. Kahn, Jocelyne M. Bray, Gerald B. Pier.

**Formal analysis:** Noah D. Cohen, Gerald B. Pier.

**Funding acquisition:** Noah D. Cohen, Colette Cywes-Bentley, Angela I. Bordin, Gerald B. Pier.

**Investigation:** Noah D. Cohen, Colette Cywes-Bentley, Susanne M. Kahn, Jocelyne M. Bray, S. Garrett Wehmeyer, Gerald B. Pier.

**Methodology:** Noah D. Cohen, Colette Cywes-Bentley, Susanne M. Kahn, Angela I. Bordin, Jocelyne M. Bray, Gerald B. Pier.

**Project administration:** Noah D. Cohen, Colette Cywes-Bentley, Angela I. Bordin, Gerald B. Pier.

**Resources:** Noah D. Cohen, Colette Cywes-Bentley, Susanne M. Kahn.

**Supervision:** Noah D. Cohen, Colette Cywes-Bentley, Angela I. Bordin, Gerald B. Pier.

**Validation:** Colette Cywes-Bentley.

**Writing – original draft:** Noah D. Cohen, Gerald B. Pier.

**Writing – review & editing:** Noah D. Cohen, Colette Cywes-Bentley, Susanne M. Kahn, Angela I. Bordin, Jocelyne M. Bray, S. Garrett Wehmeyer, Gerald B. Pier.

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
