## [Decision Letter · Decision Letter 0]

14 Sep 2020

PONE-D-20-24204

Vaccination of Yearling Horses Against Poly-N-Acetyl Glucosamine Fails to Protect Against Infection with Streptococcus equi subspecies equi

PLOS ONE

Dear Dr. Cohen,

Thank you for submitting your manuscript to PLOS ONE. After careful consideration, we feel that it has merit but does not fully meet PLOS ONE’s publication criteria as it currently stands. Therefore, we invite you to submit a revised version of the manuscript that addresses the points raised during the review process.

The two reviewers had diverse opinions on this manuscript. One thought it too preliminary and recommended "reject" the other found it clinically relevant and suggested only minor revisions. With a recommendation of "major revision" I have tried to split the difference.  In actuality, please address the comments raised by Reviewer 1 as thoroughly as possible. Upon resubmission, I will not send it back to that reviewer but strive to find another opinion. 

We look forward to receiving your revised manuscript.

Kind regards,

Nicholas J Mantis

Academic Editor

PLOS ONE

Journal Requirements:

2.  Please remove the 'DRAFT' watermark from the background of the pages in your manuscript.

"Gerald B. Pier is an inventor of intellectual properties [human monoclonal antibody to PNAG and PNAG vaccines] that are licensed by Brigham and Women’s Hospital to Alopexx Vaccine, LLC, and Alopexx Pharmaceuticals, LLC, entities in which GBP also holds equity. As an inventor of intellectual properties, GBP also has the right to receive a share of licensing-related income (royalties, fees) through Brigham and Women’s Hospital from Alopexx Pharmaceuticals, LLC, and Alopexx Vaccine, LLC. GBP’s interests were reviewed and are managed by the Brigham and Women’s Hospital and Partners Healthcare in accordance with their conflict of interest policies.

Colette Cywes-Bentley is an inventor of intellectual properties [use of human monoclonal antibody to PNAG and use of PNAG vaccines] that are licensed by Brigham and Women’s Hospital to Alopexx Pharmaceuticals, LLC. As an inventor of intellectual properties, CC-B also has the right to receive a share of licensing-related income (royalties, fees) through Brigham and Women’s Hospital from Alopexx Pharmaceuticals, LLC.

The remaining authors have declared no competing interests exist."

Reviewers' comments:

Reviewer's Responses to Questions

**Comments to the Author**

1. Is the manuscript technically sound, and do the data support the conclusions?

Reviewer #1: No

Reviewer #2: Yes

2. Has the statistical analysis been performed appropriately and rigorously? 

Reviewer #1: No

Reviewer #2: No

3. Have the authors made all data underlying the findings in their manuscript fully available?

Reviewer #1: No

Reviewer #2: Yes

4. Is the manuscript presented in an intelligible fashion and written in standard English?

Reviewer #1: Yes

Reviewer #2: Yes

5. Review Comments to the Author

Reviewer #1: In principle also negative findings should be published. However, the present study appears very preliminary and inconclusive. There are several issues not addressed.

The authors could expand a bit on why one would at all expect PNAG to be protective. Are there more examples than Rhodococcus equi? What is the variability of PNAG between different species in terms of structure, abundance and exposure.

Are there antibodies against PNAG in convalescent horses. This would show that PNAG is at all expressed by S equi and exposed to the immune system.

It is not clear if non vaccinated horses received adjuvant

The horses had a striking heterogenity of ages, 5 – 19 years.

The challenge model is questionable. Horses are constantly re-infected in this model with mingling. As the authors point out, in a normal situation an infected horse is removed from other horses to stop spreading. Has the model been used to demonstrate efficacy of other vaccines against S equi such as Pinnacle?

There are far too few horses to give conclusive results. The authors call the study a pilot study.

S equi is likely to spread via water troughs. Could S equi be recovered from these?

The authors have shown there is an immune response against PNAG. However, the opsonisation study was discontinued and lacks a negative control and is thus inconclusive. It is stated that “>50% bacteria killed at serum dilution of 10 or greater” is “good increases”. This information is meaningless if not compared with non-immunised controls or with pre-immune samples.

What do filled symbols in Fig 4 represent? Only open and half-filled are mentioned.

Reviewer #2: The topic of the study is of high clinical actuality. The paper addresses important aspect regarding vaccine development against strangles. As such the investigation is highly relevant and interesting. The manuscript is well written, however some minor revisions are recommended, as indicated below. Please see my comments below, which I believe can help at revision of the text.

Methods:

A Figure regarding methods (all four steps) with number of horses involved in each step could make it easier to follow the results and discussion.

Page 12 line 129

For each horse, the left guttural pouch was lavaged

129 with 60 ml of sterile PSS,

Lavage volume 60ml is quite low, do you think it is enough volume to get a representative sample? (In page 27 you discuss some limitations with low sampling volume, but do you think that higher volume would have been more representative?

Nasal lavage/svabs, do you think that nasal lavage could have been more representative than

General question results: Do you have any data about swabs used? t the levels of measured immunoresponse ( compared to your vaccine study in the horses that got strangles infection after exposure? )

6. PLOS authors have the option to publish the peer review history of their article (what does this mean?). If published, this will include your full peer review and any attached files.

Reviewer #1: No

Reviewer #2: No

---

## [Author Response · Author response to Decision Letter 0]

16 Sep 2020

Thank you for your electronic message dated September 14, 2020 regarding the above-referenced report. We have revised the manuscript on the basis of the comments from both reviewers. In this response letter, we detail our specific responses to comments from each of the reviewers. We hope we have satisfactorily addressed their concerns.

REVIEWER #1:

Reviewer #1: In principle also negative findings should be published. However, the present study appears very preliminary and inconclusive. There are several issues not addressed.

The authors could expand a bit on why one would at all expect PNAG to be protective. Are there more examples than Rhodococcus equi? What is the variability of PNAG between different species in terms of structure, abundance and exposure. 

AUTHORS’ RESPONSE: We thank the reviewer for the helpful suggestion to provide more information in the introduction. The introduction section of the manuscript has been revised to include additional examples of protection against other bacterial pathogens in other species. PNAG has been isolated from about 10 microbes and in all cases found by stringent chemical identification techniques to be identical, with only some variability in the level of acetylation of the amino group of the glucosamine molecule found. Published studies (Cywes-Bentley et al 2013, PNAS) document by stringent immunochemical means that the structure of PNAG is conserved among a broad range of microbes, as well as having protective efficacy against many microbes. This information has been added to the introduction of the paper. 

Are there antibodies against PNAG in convalescent horses. This would show that PNAG is at all expressed by S equi and exposed to the immune system.

AUTHORS’ RESPONSE: The manuscript has been revised to clarify that antibodies to PNAG occur naturally in many species of animals, but are usually neither opsonic nor protective because they cannot facilitate complement deposition and therefore bacterial killing. Thus, natural exposure to S. equi does not result in functional antibodies to PNAG. We do not detect antibody to PNAG in convalescent serum of horses infected with S. equi or foals infected with R. equi. 

It is not clear if non vaccinated horses received adjuvant.

AUTHORS’ RESPONSE: The manuscript has been revised to clarify that only PSS without adjuvant was administered to controls.

The horses had a striking heterogenity of ages, 5 – 19 years.

AUTHORS’ RESPONSE: The population of horses from our convenience sample of University-owned horses had a wide range of ages. Because vaccines are administered to horses of wide age ranges, we thought this was a representative population in which to examine safety and immunogenicity. 

The challenge model is questionable. Horses are constantly re-infected in this model with mingling. As the authors point out, in a normal situation an infected horse is removed from other horses to stop spreading. Has the model been used to demonstrate efficacy of other vaccines against S equi such as Pinnacle?

AUTHORS’ RESPONSE: The reviewer reiterates our point that our challenge model was strenuous. But we do not agree with the statement that there was constant re-infection. While exposure was persistent, evidence that horses infected with strangles can be re-infected during the time they have clinical signs is lacking, and evidence that infected horses are immune to infection exists. Horses were comingled, but comingling occurs naturally and horses are exposed to horses. Although horses were not removed once they developed clinical signs, treatment for strangles was implemented for these horses. We acknowledge the limitations of our model, but (disappointingly) these limitations do not vitiate the validity of our findings that the vaccine failed to protect horses. A statement was added that we cannot exclude the possibility that our vaccine would have protected against a less severe challenge.

There are far too few horses to give conclusive results. The authors call the study a pilot study.

AUTHORS’ RESPONSE: This statement is not fair. The only group in which the sample size was small was the IM+IN clinical challenge. We had more horses available for this project; however, after observing absence of evidence of protection in the 3 vaccinated horses we considered it inadvisable from a welfare perspective to expose additional horses to risk. 

S equi is likely to spread via water troughs. Could S equi be recovered from these?

AUTHORS’ RESPONSE: Environmental monitoring of exposures was beyond the scope of this study. Purulent discharge from affected horses was observed in water troughs, feed troughs, hay racks, and other locations, and we believe it is highly probable that there were viable S. equi in the water troughs, feed troughs, etc. Indeed, our assumption for the contact challenge model that these are probable sources of infection for horses.

The authors have shown there is an immune response against PNAG. However, the opsonisation study was discontinued and lacks a negative control and is thus inconclusive. It is stated that “>50% bacteria killed at serum dilution of 10 or greater” is “good increases”. This information is meaningless if not compared with non-immunised controls or with pre-immune samples.

AUTHORS’ RESPONSE: We thank the reviewer for raising this important point. The manuscript was revised to indicate that no killing of >10% was detected in comparably diluted pre-immunization samples from these horses; however, because all of the immunized horses developed clinical strangles, it was clear this measure of antibody function did not correlate with protection so we discontinued testing of the remaining samples. 

What do filled symbols in Fig 4 represent? Only open and half-filled are mentioned.

AUTHORS’ RESPONSE: We apologize for this oversight, and we have revised the manuscript to indicate that filled squares represent titers in immunized but not challenged horses. 

REVIEWER #2:

Reviewer #2: The topic of the study is of high clinical actuality. The paper addresses important aspect regarding vaccine development against strangles. As such the investigation is highly relevant and interesting. The manuscript is well written, however some minor revisions are recommended, as indicated below. Please see my comments below, which I believe can help at revision of the text.

Methods:

A Figure regarding methods (all four steps) with number of horses involved in each step could make it easier to follow the results and discussion.

AUTHORS’ RESPONSE: A schematic description has been added. I had trouble envisioning a figure that would be better than the words. We thank the reviewer for this helpful suggestion and hope the revised text serves the purpose.

Page 12 line 129

For each horse, the left guttural pouch was lavaged

129 with 60 ml of sterile PSS,

Lavage volume 60ml is quite low, do you think it is enough volume to get a representative sample? (In page 27 you discuss some limitations with low sampling volume, but do you think that higher volume would have been more representative?

AUTHORS’ RESPONSE: We understand the reviewer’s point. We chose this volume because it was one we considered would allow us to wash the walls of the pouch and collect a representative aspirate. Our concern was that further volume for lavage would have resulted in too much dilution of antibody concentrations, and would have had greater likelihood of leaking out of the pouch resulting in sample loss. 

Nasal lavage/svabs, do you think that nasal lavage could have been more representative than

General question results: Do you have any data about swabs used? t the levels of measured immunoresponse ( compared to your vaccine study in the horses that got strangles infection after exposure? )

AUTHORS’ RESPONSE: We previously found nasal lavage to yield inconsistent results for ELISAs (unpublished data). A colleague, Dr. Steeve Giguère (deceased), had similar experience with nasal lavage yielding highly variable results, and recommended using the nasal sponges based on positive experiences. We had no prior experiences, but these sponges worked well in our hands for yielding fluid that could be tested for detecting antibodies in nasal secretions.

We thank the reviewers and you for your careful consideration of our report. Please let me know if you have questions or concerns regarding this report.

---

## [Editor Report · Decision Letter 1]

28 Sep 2020

Vaccination of Yearling Horses Against Poly-N-Acetyl Glucosamine Fails to Protect Against Infection with Streptococcus equi subspecies equi

PONE-D-20-24204R1

Dear Dr. Cohen,

We’re pleased to inform you that your manuscript has been judged scientifically suitable for publication and will be formally accepted for publication once it meets all outstanding technical requirements.

Kind regards,

Nicholas J Mantis

Academic Editor

PLOS ONE